# Microbial and small zooplankton communities predict density of baleen whales in the southern California Current Ecosystem

Erin V. Satterthwaite[1]*, Trevor D. Ruiz[2], Nastassia V. Patin[1], Michaela N. Alksne[1], Len Thomas[3], Julie Dinasquet[1], Robert H. Lampe[1,4], Katherine G. Chan[2], Nicholas A. Patrick[2], Andrew E. Allen[1,4], Simone Baumann-Pickering[1], Brice X. Semmens[1]

1 Scripps Institution of Oceanography, University of California San Diego, La Jolla, California, United States of America, 2 Statistics Department, California Polytechnic State University, San Luis Obispo, California, United States of America, 3 Centre for Research into Ecological and Environmental Modelling, University of St. Andrews, St. Andrews, Fife, United Kingdom, 4 Microbial and Environmental Genomics Group, J. Craig Venter Institute, La Jolla, California, United States of America

☯ These authors contributed equally to this work.
* esatterthwaite@ucsd.edu

## Abstract

Understanding the distribution and abundance of marine mammals is important for assessing population dynamics and evaluating the impacts of human activities on these species. Here, we assessed the capability of microbial and small plankton communities to predict the density of *Balaenopteridae* whales in the Southern California Current Ecosystem in each season from 2014 to 2020 using data from the California Cooperative Oceanic Fisheries Investigations (CalCOFI). Densities of *Balaenopteridae* whales were estimated from visual line transect surveys for three target species – blue (*Balaenoptera musculus*), fin (*Balaenoptera physalus*), and humpback (*Megaptera novaeangliae*) whales – and microbial and small plankton communities were examined in concurrent water samples via metabarcoding of the 16S and 18S rRNA genes. Planktonic communities specific to each target whale species appeared as strong statistical predictors of whale estimated density, explaining 81–99% of variability and predicting density estimates to within ~1 individual per 1000 km². Our approach improved out-of-sample root mean square prediction error by up to 65% compared with simple alternative methods. Specific planktonic communities observed indicate that some predictor taxa may be ecologically associated with whales as parasites, as skin and respiratory microbiome species, or through the food chain of whale prey. However, further studies are needed to understand how these organisms function collectively as a community and interact with the "ecological habitat" that supports whales. Our results suggest that using planktonic communities to quantify the potential ecological habitat of larger organisms, like baleen whales, can enhance predictive models and may inform hypotheses about the ecological relationships between whales and the biological communities with which they co-occur.

**Data availability statement:** Sequence data generated in this study have been deposited in the NCBI Sequence Read Archive under BioProject accession numbers PRJNA555783, PRJNA665326, and PRJNA804265. Sighting data, density estimates, NCOG sample metadata, taxonomic annotations, and data used for modeling are publicly available via Zenodo at https://doi.org/10.5281/zenodo.15678927. R code to reproduce the analyses is available on GitHub at https://github.com/ruizt/marine-mammal-edna.

**Funding:** This material is based upon research supported by: Office of Naval Research (N00014-22-1-2719 to EVS, SBP, BXS, and LT); National Oceanic and Atmospheric Administration (NA15OAR4320071 and NA19NOS4780181 to AEA); Simons Foundation Collaboration on Principles of Microbial Ecosystems (PriME) (970820 to AEA); US Navy Pacific Fleet (N62473-18-2-0016, N62473-19-2-0028, and N62473-16-2-0012 to SBP); and the Research, Scholarly & Creative Activities Program awarded by the Cal Poly Division of Research (to TDR). Additionally, EVS was supported by a partnership among CalCOFI participants, including Scripps Institution of Oceanography (SIO), NOAA Southwest Fisheries Science Center (NA20OAR4170258), California Department of Fish and Wildlife (P2370002), and California Sea Grant (NA22OAR4170106). The funders had no role in study design, data collection and analysis, decision to publish, or preparation of the manuscript.

**Competing interests:** The authors have declared that no competing interests exist.

## Introduction

Large baleen whales play a vital role in marine ecosystems by helping to regulate ecosystem processes [1]. They are of conservation and management relevance as many populations are listed as threatened (Vulnerable, Endangered, or Critically Endangered) on the IUCN Red List of Threatened Species [2], and they provide significant cultural value to people [3]. In the eastern North Pacific, blue (*Balaenoptera musculus*), fin (*Balaenoptera physalus*), and humpback (*Megaptera novaeangliae*) whales are among the most widespread and frequently encountered of the family *Balaenopteridae*, with long-range migrations that connect low-latitude wintering grounds to mid- and high-latitude summer foraging grounds [4,5]. Each year, blue, fin, and humpback whales migrate to the productive waters of the California Current to feed on dense aggregations of krill and schooling fishes [6–8]. Their foraging grounds and migratory corridors overlap with commercial shipping routes and areas of military activity, particularly in Southern California [9,10], making the ability to predict baleen whale distribution an important management directive in the region.

The seasonal distribution and abundance of these species in the southern California Current Ecosystem (CCE) is largely shaped by oceanographic conditions that influence prey availability. Blue whales preferentially consume specific krill species, favoring *Thysanoessa spinifera* over *Euphausia pacifica* [11], and their distributions tend to be correlated with krill aggregations [6,12]. Humpback whales are opportunistic foragers, feeding on both krill and small schooling fish such as sardine, anchovy, sand lance, and herring [13–15], with diet composition likely reflecting prey community structure and underlying oceanographic conditions [7]. Fin whales are also opportunistic foragers, though less is known about their prey preferences, seasonal distribution, and general ecology; they occur year-round in the CCE [16] but, like blue whales, peak density occurs in summer months [17]. Across species, migratory phenology and interannual variability are closely linked to environmental conditions. For instance, long-term acoustic studies have documented shifts in call occurrence associated with biological productivity [18,19], sea surface temperature [20], and broader events such as marine heatwaves and decadal-scale climate fluctuations [21].

The close association between baleen whales and their preferred habitat suggests that their density in the CCE is linked to habitat-specific ecological characteristics [22]. We use the term "ecological habitat" to describe the community of small organisms whose presence reflects the ecological conditions that support the occurrence of another larger organism (following [23]). The ecological habitat of whales includes taxa linked to them through food-web pathways, such as bacteria, protists, and small zooplankton that contribute to productivity or serve as prey of whale prey. We also include taxa that may be directly or indirectly biologically associated such as whale parasites, commensal organisms, and microorganisms found on whale skin [24–30]; in their gut [31,32]; or in respiratory fluids [33,34]. Whale microbiomes, which consist of communities of microorganisms living on and in the whale, have been found to significantly differ from the microbial community in the surrounding seawater [33,35], and distinct bacterial taxa have been associated with the skin of humpback whales

across the North Pacific [29]. Additional studies have observed parasites, viruses, and epibiotic fauna specific to whales [28].

Beyond studies of microbial and other taxa living on or in whales, previous research has linked prey and zooplankton biomass or abundance to baleen whale distribution [36,37], and recent work has used molecular methods to characterize baleen whale prey species [38,39]. Thus, the ecological habitat of baleen whales is not limited to potential zooplankton prey but also includes prokaryotic microbes, small eukaryotic heterotrophic microbes, phytoplankton, and protozoans.

Understanding these ecological relationships complements efforts to monitor the distributions of baleen whales over time, which is necessary for tracking population changes and understanding the effects of human activity on these species. Baleen whales are monitored using a combination of visual surveys and photo identification [17,40,41], acoustic methods [8,16], satellite imagery [42], tags [43], and, increasingly, genetic methods [44]. Sampling whales and other marine mammals is challenging due to their wide-ranging and often patchy distributions [45], low encounter rates [17], brief or intermittent surface cues [46], deep-diving or cryptic behaviors [47], and the high logistical, financial, and permitting costs of direct or invasive sampling methods.

Because large baleen whales are difficult to sample directly, yet are closely linked to their habitat and associated biota, we investigated whether microbial and small plankton communities could serve as proxies for predicting their density in the southern California Current Ecosystem. The California Current Ecosystem was chosen because it is a highly productive upwelling system and is known to be an important foraging ground for baleen whales [9,10].

We leveraged six years of ship-based marine mammal data coupled with data on the microbial and small-plankton community (hereafter planktonic communities) to predict seasonal and interannual baleen whale density. We used environmental DNA (eDNA) metabarcoding to characterize baleen whale ecological habitat and to predict their seasonal density in the California Current Ecosystem. This approach captures DNA from single-celled microbes and cells shed by multicellular organisms into the water column to generate a molecular "fingerprint" of the biological community spanning multiple trophic levels [48,49]. We used two well-established genetic markers, the 16S and 18S ribosomal RNA genes, to capture a wide range of prokaryotic and eukaryotic microbes as well as small metazoan zooplankton like copepods and krill [50,51]. We focus our analyses on blue, fin, and humpback whales, given that they are abundant in the California Current Ecosystem [17,52], forage at low trophic levels [12], and have been shown to have existing connections to various microbes [29,31]. Additionally, concurrent eDNA samples and visual sightings of baleen whales exist from the California Cooperative Oceanic Fisheries Investigations (CalCOFI), the longest integrated marine ecosystem observing program in the world. This approach allows us to evaluate the extent to which marine planktonic communities can serve as predictors of the ecological habitats of top consumers, like baleen whales. By predicting seasonal and interannual baleen whale density from microbial community composition, this work may help quantify broad ecological linkages between baleen whales and microbes on a key foraging ground, potentially guiding the inclusion of such community-level associations into future habitat suitability and species distribution modeling efforts.

## Methods

### Sampling area

The Southern California Bight region is situated in the southern portion of the California Current Ecosystem (CCE), a productive upwelling system that supports important baleen whale species. The California Cooperative Oceanic Fisheries Investigations (CalCOFI), one of the longest running integrated marine ecosystem monitoring programs in the world, has systematically sampled the physics, chemistry, and biology of the CCE since 1949.

Quarterly CalCOFI sampling consists of 75 stations along six "core area" transects that extend from San Diego, CA to north of Point Conception (Morro Bay, CA) and include coastal stations (~50 m depth) and stations within the core of the California Current (CC) out to ~250–550 km offshore. The transects are spaced approximately 40 nautical miles (nm) apart. Along the transect lines, stations are spaced approximately 40 nm apart for offshore stations and 20 nm apart for

coastal stations. In this project we utilize data collected from the core sampling area comprising stations located on Cal-COFI lines 93.3 to 76.7 (SE corner: 32.956, −117.305; NE corner: 35.088, −120.777; NW corner: 33.388, −124.323; SW corner: 29.846, −123.587) between 2014 and 2020.

## Environmental DNA collection and amplicon sequencing

From select stations on CalCOFI cruises, 743 DNA samples were collected within the core sampling area from 2014−2020 as part of the NOAA-CalCOFI Ocean Genomics (NCOG) time series [53]. We sampled key stations on lines 80 and 90, as well as basin stations, to provide an onshore-offshore gradient across two transects. Seawater was collected from the near-surface (normally 10 m) and the subsurface chlorophyll maximum layer and filtered onto 0.22 µm Sterivex™ filters (MilliporeSigma SVGP01050) that were immediately flash frozen in liquid nitrogen and stored at −80°C. In between samples, bottles and lines were rinsed with Milli-Q water, then rinsed again three times with the sample itself prior to beginning filtration. The average volume filtered was 3.3 L. Following each cruise, samples were brought to J. Craig Venter Institute for processing with equipment and workspaces cleaned with 70% ethanol between each use. DNA was extracted with the Macherey-Nagel NucleoMag Plant kit (Cat. no. 744400) on an Eppendorf epMotion 5075TMX and assessed on a 1.8% agarose gel. Blank samples were included during extraction to confirm the absence of contamination.

Amplicon libraries separately targeting the V4-V5 region of the 16S rRNA gene and both the V4 and V9 regions of the 18S rRNA genes were constructed via a one-step PCR with the TruFi DNA Polymerase PCR kit (Cat. no. AZ-1702). The primers used here were established by previous studies to capture the entire microbial community with commonly-used marker gene regions, and have been shown to outperform other primer sets [51]. For 16S, the 515F-Y (5′-GTG YCA GCM GCC GCG GTA A-3′) and 926R (5′-CCG YCA ATT YMT TTR AGT TT-3′) primer set was used [54]. For 18S-V4, the V4F (5′-CCA GCA SCY GCG GTA ATT CC-3′) and V4RB (5′-ACT TTC GTT CTT GAT YR-3′) primer set modified from [55] was used. For 18S-V9, the 1389F (5′-TTG TAC ACA CCG CCC-3′) and 1510R (5′-CCT TCY GCA GGT TCA CCT AC-5′) primer set was used [56]. In addition to the extraction blank samples, negative PCR controls were also included.

Each reaction was performed with an initial denaturing step at 95°C for 1 minute followed by 30 cycles of 95°C for 15 seconds, 56°C for 15 seconds, and 72°C for 30 seconds. 2.5 µL of each PCR reaction was run on a 1.8% agarose gel to confirm amplification, then PCR products were purified with Beckman Coulter AMPure XP beads (1x) following the manufacturer's instructions. DNA quantification of the PCR products was performed in duplicate using the Invitrogen Quant-iT PicoGreen dsDNA Assay kit (Cat. no. P7589). All samples regardless of positive amplification as well as a subset of extraction blanks and PCR controls were then combined in equal proportions where possible into multiple pools followed by another 0.8x AMPure XP bead purification on the final pool. DNA quality of each pool was evaluated on an Agilent 2200 TapeStation, and quantification was performed with the Invitrogen Qubit HS dsDNA kit (Cat. no. Q32854). Each 16S or 18S pool was sequenced on an Illumina MiSeq (2 x 300 bp for 16S and V4 or 2 x 150 bp for V9) except for the one pool for the 2014–2016 euphotic zone V9 samples, which was run on an Illumina NextSeq 500 (Mid Output, 2 x 150 bp).

Amplicons were analyzed with QIIME2 v2019.10 [57]. Briefly, paired-end reads were trimmed to remove adapter and primer sequences with cutadapt [58]. Trimmed reads were then denoised with DADA2 to produce amplicon sequence variants (ASVs). Each run was denoised with DADA2 separately to account for different error profiles in each run then merged. Taxonomic annotation of ASVs was performed with the q2-feature-classifier naïve bayes classifier using the SILVA database (Release 138) for 16S ASVs and the PR$^2$ database (v4.13.0) for 18S ASVs [59–62].

## Visual surveys of baleen whales

Since 2004, visual sightings of baleen whales have been recorded during cruises along CalCOFI transects. In this project, we focus on data from 2014 to 2020 collected contemporaneously with the NCOG data described in the previous section (Environmental DNA collection and amplicon sequencing). The dataset includes marine mammal monitoring effort from 25 individual CalCOFI cruises spanning all four seasons and limited to the core sampling area.

 

Visual monitoring effort was conducted in "passing mode" and adapted from standard line transect marine mammal survey protocols [63,64] following methods outlined in [17]. Two trained marine mammal observers used 7x50 Fujinon binoculars to observe and record marine mammals during daylight hours as the ship transited between CalCOFI stations. Observers systematically recorded species identification, group size estimates, reticle position below the horizon, angle relative to the bow, latitude and longitude, ship's heading, sea state, swell height and visibility. Survey effort was suspended when sea state was greater than Beaufort 6 or when visibility less than 1 km.

Whale sightings were only included when classified as both "on-effort" and "on-transect". The on-effort criteria was met when two observers were actively scanning while the vessel was traveling above 10 knots in a sea state below Beaufort 6 with greater than or equal to 1 km visibility. The on-transect criteria was met when sightings were along one of the core CalCOFI transect lines.

## Estimated baleen whale densities

The marine mammal visual survey data was used to estimate density (number of individuals per 1000 km$^2$) using multiple covariate distance sampling methods [65]. This analysis involved two stages: (1) estimating each species' detectability as a function of factors potentially affecting sighting conditions; (2) estimating species' density per cruise given the number detected and the estimated detectability. The major advantage of this approach over simply using sighting rates (number of individuals detected per unit survey effort) is that it can account for differences in sighting rates that are caused by differences in detectability (for example, if sighting conditions tend to be worse in winter) that might otherwise confound downstream inferences about the relationship between whales and their ecological habitat (see, e.g., [66]). Distance sampling analyses were undertaken using the Distance R package [67].

Distance sampling methods for line transect surveys use the distribution of perpendicular distances of observed animals to estimate a "detection function" (i.e., probability of detection as a function of perpendicular distance and other covariates), and from this the average probability of detection within the surveyed strips [68]. For each sighting, the measured reticle position and angle relative to bow were used, together with the knowledge of observer eye height above the water, to calculate perpendicular distance for each sighting. Following [17], a perpendicular truncation distance of 2400 m was used. For each species, candidate detection functions were fitted incrementally using forward selection, starting with a key function and adding terms if the resulting model had a lower Akaike Information Criterion (AIC) score. Key functions were uniform, half-normal and hazard rate. In one set of analyses the terms added were cosine (with uniform and half-normal) or polynomial (with hazard rate) adjustment terms. In another set of analyses using just half-normal and hazard-rate key functions, the terms added were covariates affecting the scale parameter of the key functions: Beaufort sea state, swell height, observer height above water (as a factor with 3 levels) and group size. The final model chosen for inference was the one with lowest AIC over both sets. A list of all candidate models is given in supporting information (S5 Table). Goodness of fit was assessed visually by comparing the fitted detection function with histograms of observed distances, and using a Cramér-von Mises test.

Given a fitted detection function, estimated density per cruise, denoted $y_i$, for each species was calculated as:

$$y_i = \frac{1}{2L_i w} \sum_{g=1}^{n_i} \frac{s_g}{\hat{p}_g(z_g)}$$

where $L_i$ is the transect length on cruise $i$, $w$ is the truncation distance, $n_i$ is the number of detections of the species on survey $i$, $s_g$ is the group size of the $g$th detection, and $\hat{p}_g(z_g)$ is the estimated detection probability of the $g$th detection given its covariates $z_g$. This detection probability is computed by averaging the detection function over the perpendicular distances:

$$\hat{p}_g(z_g) = \frac{1}{w} \int_{x=0}^{w} \hat{g}(x, z_g)dx$$

where $\hat{g}(x, z_g)$ is the estimated detection function and $x$ is perpendicular distance. Variance in estimated density was calculated by combining variance in the estimated detection function and variance in detection rate between transect lines, as detailed by [69].

## Analysis of amplicon relative abundances

We removed rare ASVs present in under 1% of samples across all cruises and abundant ASVs present in more than 99% of samples across all cruises, as neither rare nor ubiquitous ASVs yield data with sufficient variation to provide a basis for prediction of marine mammal density. Under the assumption that all remaining ASVs were physically present across samples and cruises, geometric Bayesian multiplicative count zero imputation [70] was used to estimate relative abundances for non-detections.

To match the spatial resolution of the whale density estimates, amplicon relative abundances were aggregated to the cruise level by weighted (geometric) averaging. In detail, if $x_{ijklm}$ denotes the relative abundance of ASV $j$ from the sample taken at station $l$ on transect $k$ and depth $m$ on cruise $i$, relative abundances were aggregated across depth and sampling location by taking a weighted geometric mean:

$$x_{ij} = \prod_{k=1}^{K_i} \prod_{l=1}^{L_{ik}} \prod_{m=1}^{2} \left(x_{ijklm}\right)^{w_{klm}}$$

Weights $w_{klm}$ were inversely proportional to spatial sampling density with respect to geolocation and maximized $\alpha$-diversity with respect to depth; the latter criterion resulted in weights slightly favoring samples taken at max chlorophyll-$a$ depth. The resulting quantity measures the average relative abundance of ASV $j$ observed across samples collected on cruise $i$.

The centered log-ratio (CLR) transformation [71] was then applied to average relative abundances $x_{ij}$. The "typical" average relative abundance taken across ASVs on cruise $i$ is given by the geometric mean:

$$g_i = \left(\prod_{j=1}^{J} x_{ij}\right)^{1/J}$$

The CLR transformation is defined as the natural logarithm of the ratio:

$$z_{ij} = x_{ij}/g_i$$

This captures the factor by which the average relative abundance of a particular ASV deviates from the typical average relative abundance across all ASVs on a given cruise; for example, value of $z_{ij} = 2$ indicates that ASV $j$ is twice as abundant as the typical ASV on cruise $i$.

The above aggregations and transformations were performed separately for the 16S, 18S-V4, and 18S-V9 markers, yielding three sets of $z_{ij}$; this level of detail is omitted from the notation.

## Seasonal log-ratios

Seasonality was removed from the whale density and amplicon data with a secondary log-ratio transformation using the respective seasonal averages. The seasonal geometric means, written as functions of the observation index (cruise) $i$, are:

$$g(i, y) = \left( \prod_{i \in I(i)} y_i \right)^{1/|I(i)|}$$

$$g(i, z_j) = \left( \prod_{i \in I(i)} z_{ij} \right)^{1/|I(i)|}$$

In these expressions $I(i)$ is an index set comprising the indices of all observations made in the same quarter as observation $i$. The seasonally-adjusted estimated whale density and the seasonally-adjusted average amplicon relative abundance are then $y_i/g(i, y)$ and $z_{ij}/g(i, z_j)$, respectively. The resulting quantities are best interpreted as deviations from seasonal averages; for example, a value of $y_1/g(1, y) = 0.5$ would indicate that on the first cruise, observed density was half of the seasonal average for the corresponding quarter.

## Log-contrast model framework

We formulated a log-contrast-type model [72] to identify and estimate statistical relationships. This model expresses the seasonally-adjusted estimated densities as linear functions of the seasonally-adjusted average amplicon relative abundances:

$$log\left( \frac{y_i}{g(i, y)} \right) = \beta_0 + \sum_{j=1}^{J} \beta_j \cdot log\left( \frac{z_{ij}}{g(i, z_j)} \right) + \epsilon_i$$

(1)

Due to the logratio transformations, the coefficients capture multiplicative changes in median estimated density associated with multiplicative changes in relative abundances, after adjusting for seasonality and assuming error normality. For example, a twofold change in the relative abundance of ASV $j$, relative to its seasonal average, is associated with a change in median density, relative to its seasonal average, of a factor of $2^{\beta_j}$. We specified separate models for each whale species of interest and each marker, amounting to 9 models in total.

## Variable selection and parameter estimation

A partial least squares (PLS) latent variable framework [73,74] was used for variable selection and parameter estimation. PLS allows for estimation of the full set of model coefficients even when least squares is ill-posed due to the number of covariates (ASVs) exceeding the number of samples (cruises) [75]. Writing the log-contrast model (Eq. 1) in linear model form $Y = Z\beta + \epsilon$, the PLS framework stipulates a set of latent variables or "components":

$$T = ZA$$

The columns $a_k$ of $A$ or component "loadings" are estimated sequentially by component $k = 1, \ldots, K$ by maximizing the correlation with the response and the variance of the latent component, subject to an orthogonality constraint with respect to previous latent components and a unit-norm constraint (when $k = 1$ the orthogonality constraint is omitted):

$$a_k = argmax_a \left\{ corr^2(Za, Y)var(Za) \right\} \text{ subject to } a \perp \Sigma_z a_{k-1} \text{ and } a^T a = 1$$

(2)

The SIMPLS algorithm [76] was used to compute estimates of the loading matrix $A$. Subsequently, a linear model was fit with the latent components $T$ as covariates and least squares estimates were back-propagated to obtain coefficient estimates for the model as originally specified:

$$\beta^* = A(T'T)^{-1}T'Y$$

The log-contrast models (Eq. 1) were specified using a small subset of candidate amplicons identified through variable selection to improve model interpretability and prioritize identification of relatively stronger associations. The selection procedure resulted from applying the stability selection method of [77] to sparse partial least squares (sPLS) estimates of $A$ [78]. The sPLS method introduces an $L_1$ penalty to the PLS optimization problem (Eq. 2), which has the effect of shrinking small component loadings $a_{kj}$ to exactly zero and inducing sparsity in the loading matrix $A$. In [78], the authors approximate the solution to the resulting problem using a surrogate approach with a hyperparameter $\lambda$ controlling the degree of sparsity in the sPLS estimate; this results in a sparse coefficient estimate $\beta^\lambda$. Stability selection is a computationally-intensive procedure that traverses the problem of hyperparameter tuning by estimating the selection probability of each variable for a "path" of hyperparameter values in a specified region $\Lambda$. Selection probability estimates are obtained by computing $\beta^\lambda$ repeatedly from subsamples of the data.

We used leave-one-out partitions to estimate, for each hyperparameter $\lambda \in \Lambda$ and each candidate amplicon $j$, the probability of selecting that amplicon using the sPLS method:

$$\pi_j^\lambda = P\left(\beta_j^\lambda \neq 0\right)$$

The "stable set" consists of frequently-selected amplicons, specifically those variables whose estimated selection probability exceeds $\pi_{max}$ for at least one $\lambda \in \Lambda$:

$$S_\Lambda = \left\{ j : \pi_j^\lambda \geq \pi_{max} \ \text{for some} \ \lambda \in \Lambda \right\} \tag{3}$$

In [77], the authors provide heuristics for choosing the region $\Lambda$ to control the expected number of falsely selected variables (per-family error rate); we determined $\Lambda$ according to their method in order to control the per-family error rate at 0.5.

In the context of our model, the sPLS coefficient estimate $\beta^\lambda$ depends not only on the sparsity hyperparameter $\lambda$ but also on the number $K$ of latent components. We addressed this by estimating stable sets $S_\Lambda^K$ for $K = 4, ..., 12$ and choosing the number of components that optimized mean square prediction error estimated from leave-one-out partitions of the data. We re-computed seasonal adjustments when forming data partitions so that subsamples did not incorporate information about held-out observations via seasonal averages. Once the stable set $S$ was estimated for each model, we computed SIMPLS estimates of the model coefficients with the number of latent components $K$ used to determine the stable set.

## Model validation

We sought to further assess the consistency of the variable selection procedure by comparing stable sets obtained under perturbations of the data partitions used to estimate selection probabilities. An "outer validation" was performed by constructing a set of nested leave-one-out partitions and performing the entire stability selection procedure holding out one observation (cruise) at a time. This yielded stable sets $S_1, ..., S_n$ (one from holding out each of the $n = 25$ cruises) which we then compared for consistency using a thresholded Jaccard index:

$$J_\alpha = \frac{\left|\left\{j : \sum_{i=1}^n 1\left\{j \in S_i\right\} \geq n\alpha\right\}\right|}{\left|\cup_{i=1}^n S_i\right|} \tag{4}$$

This measures the proportion of amplicons selected at least $\alpha$% of the time among the stable sets obtained from the outer validation procedure. We chose $\alpha = 0.5$ to look at the proportion of amplicons selected more often than not across validation runs as a measure of consistency of the variable selection procedure.

**Narrative review of direct relationships between baleen whales and microbes/small zooplankton**

A narrative review was conducted to identify known direct relationships between blue, humpback, and fin whales and bacteria, microbes, and small plankton from the existing literature for comparison with potential relationships identified by our models. The review focused on studies that examine microbial and planktonic interactions with these baleen whale species. The Publish or Perish software was utilized to search the academic literature database Google Scholar for peer-reviewed articles and reports. Specific title-keyword combinations were searched to capture relevant studies related to bacteria, microbes, and plankton associated with blue, humpback, and fin whales. Search terms included the common name of each whale species in the title and "bacteria," "plankton," or "microbes" in the keywords.

The titles of these papers were reviewed to assess relevance to the research question, focusing on studies that explored direct connections between baleen whales and microbes, bacteria, or plankton. Studies that examined direct interactions between microbes or small plankton and baleen whales were selected. Interactions included topics such as: baleen whale health and disease, including microbial diversity and baleen whale pathology; feeding ecology, including studies that analyzed prey-plankton dynamics and their relationship with baleen whale foraging behavior; baleen whale microbiome, such as studies on respiratory, gut, or skin microbiomes; baleen whale parasitology and pathology, which examined diseases and parasites associated with baleen whale health; and baleen whale strandings and carcasses. Studies unrelated to direct baleen whale-microbe relationships were excluded from the review. This search strategy yielded 18 relevant papers.

These selected papers were examined for relevant information that was entered into a structured table that contained the following information: citation, title of the paper, url/link to the paper, notes, whale species (e.g., blue, humpback, or fin whales), type of relationship between the whale species and bacteria, microbes, or plankton, type of sample or data used in the study, location, method, taxonomic classification of the bacteria/microbe/small plankton (Kingdom/Domain, Phylum, Class, Order, Infraorder, Family, Genus, Accepted Name).

For each microbial taxon mentioned in the selected studies, a detailed taxonomic classification was retrieved programmatically in Python using the World Register of Marine Species (WoRMS) API and programmatically in R using the NCBI (National Center for Biotechnology Information) database (S4 Table).

## Results

Qualitative assessment of microbial community data, baleen whale sightings, and density estimates identified spatial and temporal patterns in both baleen whale sighting rates and density estimates across species, and relatively homogeneous sampling across space and time suggests minimal bias in measurement of microbial communities (Section *Datasets*). Statistical models relating baleen whale density estimates to microbial community data identified small sets of ASVs that explained a large proportion of variation in whale density estimates after adjusting for seasonality (Section *Estimated Relationships*). The relative abundances of taxa within these subcommunities provide accurate predictions of density of target baleen whale species, and community predictors outperform naive forecasting methods (Section *Model Predictions*). Through a narrative review of existing literature, we found some of the taxa in our study have also been previously documented as microbial associates of baleen whales (Section *Communities of Taxonomic Annotations*). Lastly, despite some taxonomic overlap in microbial subcommunity predictors of blue, humpback, and fin whales, our models suggest that each species is related to a distinct subcommunity (Section *Narrative Review Findings*).

### Datasets

Fig 1 shows the locations along CalCOFI survey transects of (a) NCOG samples sequenced and (b) on-effort whale sightings used in the analysis; sightings tended to occur nearer to shore. The distribution of NCOG samples across space and time was approximately uniform (Table 1); data for a given cruise and genetic marker typically comprised 20–40 samples collected across 5–6 transects during a 2–3 week period. However, there is some variation in the number of samples

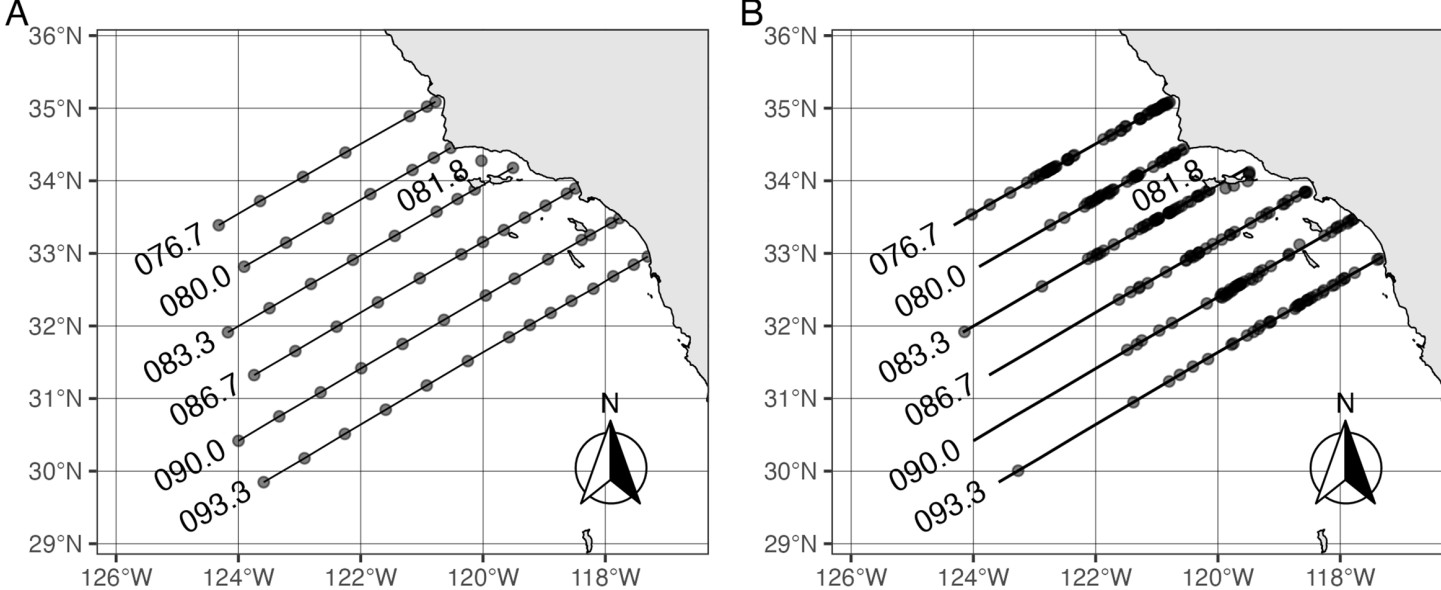

**Fig 1. Locations of NCOG samples and baleen whale sightings along CalCOFI transects. (A)** Sampling locations for NCOG samples used in the analysis and **(B)** sighting locations recorded from visual survey data for target species during 2014-2020. All on-effort, on-transect sightings are shown; in the line transect analysis, the small number of sightings at perpendicular distances greater than 2400 m were truncated.

sequenced by cruise – with as few as 14 samples collected in spring 2014 and winter 2019 and as many as 56 collected in winter 2018 – as well as in the length of the survey period and the spatial distribution of sampling locations.

S1 Table shows NCOG sample counts by marker, cruise, and transect, providing a more fine-grained look at the spatial distribution of these samples. Certain transects – namely, 080.0, 090.0, and 093.3 – were consistently sampled more densely. S2 Table lists the unique ASVs retained in the analysis after filtering out rare and ubiquitous ASVs for the 16S, 18S-V4, and 18S-V9 markers, along with their taxonomic classifications, if known. There were 6234, 6824, and 9511 such "candidates", respectively. Known taxonomic classifications among candidates (*i.e.,* after relative abundance filtering) exhibited relatively less overlap between 16S and 18S markers (9.9% between 16S and 18S-V9 and 0.7% between 16S and 18S-V4 at the genus level) than between the two 18S markers (51.2% between 18S-V4 and 18S-V9 at the genus level).

Whale sightings exhibited clear seasonal variation (Table 2): blue whales were the most seasonal and rarely sighted except in summer; fin whales were also more common in summer, but sighted throughout the year in most years; humpback whales were sighted year-round but in greater numbers during spring. Sample sizes for the distance sampling analysis, and tables showing AIC values for fitted detection function models are given in S5 Table. The selected (i.e., lowest-AIC) models were uniform with one cosine adjustment for blue whales, half-normal with sea state, group size and observer platform height as covariates for fin whales and half-normal with swell, platform height and swell as covariates for humpback whales. Selected models for all 3 species were good fits to the distance data (Craemer von-Mises test p-values 0.70, 0.31, and 0.59 respectively). Estimated density varied by season in a similar manner to the sighting rates, with strong inter-annual variability (Fig 2).

### Estimated relationships

Our analysis identified sparse well-fitting models, selecting stable sets (Eq. 2) comprising between 23 and 60 ASVs depending on marker and whale species (between 0.24% and 0.85% of candidates) that explain an estimated 81–99% of variation in density estimates after adjusting for seasonality (Table 3). Residual diagnostic checks indicated no issues with model

**Table 1. Counts of NCOG samples and CalCOFI transects per cruise by marker.**

| Year | Season | Sampling start date | Sampling end date | Samples (16S) | Transects (16S) | Samples (18S-V4) | Transects (18S-V4) | Samples (18S-V9) | Transects (18S-V9) |
|---|---|---|---|---|---|---|---|---|---|
| 2014 | winter | 2014-01-29 | 2014-02-04 | 14 | 2 | 12 | 2 | 12 | 2 |
| 2014 | spring | 2014-04-02 | 2014-04-16 | 32 | 6 | 32 | 6 | 32 | 6 |
| 2014 | summer | 2014-07-06 | 2014-07-19 | 26 | 4 | 26 | 4 | 24 | 4 |
| 2014 | fall | 2014-11-08 | 2014-11-22 | 26 | 6 | 24 | 6 | 24 | 6 |
| 2015 | winter | 2015-01-15 | 2015-01-29 | 28 | 7 | 28 | 7 | 28 | 7 |
| 2015 | spring | 2015-04-08 | 2015-04-17 | 14 | 2 | 14 | 2 | 14 | 2 |
| 2015 | summer | 2015-07-08 | 2015-07-23 | 24 | 6 | 22 | 6 | 24 | 6 |
| 2015 | fall | 2015-11-01 | 2015-11-10 | 22 | 5 | 22 | 5 | 22 | 5 |
| 2016 | winter | 2016-01-09 | 2016-01-20 | 22 | 6 | 22 | 6 | 22 | 6 |
| 2016 | spring | 2016-04-01 | 2016-04-14 | 22 | 6 | 22 | 6 | 22 | 6 |
| 2016 | summer | 2016-07-10 | 2016-07-24 | 18 | 5 | 18 | 5 | 18 | 5 |
| 2016 | fall | 2016-11-06 | 2016-11-20 | 26 | 6 | 26 | 6 | 26 | 6 |
| 2017 | winter | 2017-01-06 | 2017-01-18 | 38 | 7 | 32 | 6 | 38 | 7 |
| 2017 | spring | 2017-03-29 | 2017-04-12 | 32 | 6 | 32 | 6 | 32 | 6 |
| 2017 | summer | 2017-08-02 | 2017-08-13 | 34 | 5 | 26 | 4 | 34 | 5 |
| 2017 | fall | 2017-11-09 | 2017-11-22 | 32 | 5 | 20 | 4 | 32 | 5 |
| 2018 | winter | 2018-02-01 | 2018-02-09 | 56 | 6 | 56 | 6 | 56 | 6 |
| 2018 | spring | 2018-04-06 | 2018-04-20 | 32 | 6 | 32 | 6 | 30 | 6 |
| 2018 | summer | 2018-06-11 | 2018-06-23 | 32 | 7 | 32 | 7 | 32 | 7 |
| 2018 | fall | 2018-10-14 | 2018-10-27 | 34 | 6 | 34 | 6 | 34 | 6 |
| 2019 | winter | 2019-02-07 | 2019-02-12 | 14 | 4 | 14 | 4 | 14 | 4 |
| 2019 | spring | 2019-04-03 | 2019-04-17 | 32 | 7 | 32 | 7 | 32 | 7 |
| 2019 | summer | 2019-07-11 | 2019-07-26 | 44 | 7 | 44 | 7 | 44 | 7 |
| 2019 | fall | 2019-11-05 | 2019-11-17 | 34 | 7 | 36 | 7 | 36 | 7 |
| 2020 | winter | 2020-01-05 | 2020-01-18 | 26 | 6 | 26 | 6 | 26 | 6 |
| 2020 | spring | 2020-07-14 | 2020-07-25 | 26 | 6 | 26 | 6 | 26 | 6 |
| 2020 | summer | 2020-10-12 | 2020-10-22 | 32 | 6 | 32 | 6 | 32 | 6 |

Counts reflect the samples used in the analysis rather than the total number of samples collected during the survey; similarly, start and end dates indicate the earliest and latest dates of samples included in the analysis.

specification or time dependence between successive surveys (S1 Fig). Optimal model hyperparameters varied slightly; models used between 4 and 11 latent components (a maximum of 12 were available for hyperparameter optimization). The selected ASVs spanned 7–19 classes, 8–25 orders, and 9–28 families, again depending on target species and marker.

Our model selection procedure exhibited some sensitivity to data perturbation. Depending on the model and taxonomic level, anywhere from 22–68% of selected taxa are robust to leave-one-out data perturbations. In detail, Table 4 shows the modified Jaccard index (Eq. 4) computed for each model at the ASV, family, order, and class levels. This measure quantifies the proportion of taxa enumerated across models fit to leave-one-out data partitions that are included more often than not in the stable set.

S3 Table enumerates selected ASVs by whale species and marker (*i.e.,* by model) along with taxonomic classifications, if known, and an estimated measure of association (*i.e.,* estimated model coefficient in Eq. 1) quantifying multiplicative change in whale sightings per doubling of ASV relative abundance after adjusting for seasonality. For example, an estimate of 1.338 would indicate that every doubling of the relative abundance of that particular microbe (relative to its seasonal average) is associated with an estimated 33.8% increase in estimated blue whale sightings (relative to its

**Table 2. Number of whale sightings per cruise among whale species of interest.**

| Year | Season | Survey start | Survey end | Fin whale sightings | Humpback whale sightings | Blue whale sightings |
|---|---|---|---|---|---|---|
| 2014 | winter | 2014-01-29 | 2014-02-04 | 2 | 0 | 0 |
| | spring | 2014-04-03 | 2014-04-15 | 3 | 13 | 0 |
| | summer | 2014-07-06 | 2014-07-21 | 17 | 2 | 6 |
| | fall | 2014-11-09 | 2014-11-22 | 1 | 8 | 1 |
| 2015 | winter | 2015-01-16 | 2015-01-29 | 0 | 5 | 0 |
| | spring | 2015-04-05 | 2015-04-18 | 0 | 4 | 0 |
| | summer | 2015-07-08 | 2015-07-24 | 12 | 0 | 9 |
| | fall | 2015-10-29 | 2015-11-11 | 0 | 4 | 0 |
| 2016 | winter | 2016-01-08 | 2016-01-21 | 1 | 2 | 0 |
| | spring | 2016-04-02 | 2016-04-14 | 3 | 3 | 3 |
| | summer | 2016-07-11 | 2016-07-25 | 3 | 1 | 10 |
| | fall | 2016-11-07 | 2016-11-20 | 1 | 2 | 0 |
| 2017 | winter | 2017-01-06 | 2017-01-19 | 1 | 4 | 1 |
| | spring | 2017-03-29 | 2017-04-12 | 0 | 6 | 0 |
| | summer | 2017-08-01 | 2017-08-13 | 4 | 1 | 10 |
| | fall | 2017-11-10 | 2017-11-23 | 3 | 4 | 0 |
| 2018 | winter | 2018-02-02 | 2018-02-09 | 0 | 3 | 0 |
| | spring | 2018-04-06 | 2018-04-20 | 2 | 9 | 0 |
| | summer | 2018-06-09 | 2018-06-22 | 4 | 3 | 2 |
| | fall | 2018-10-15 | 2018-10-27 | 5 | 4 | 0 |
| 2019 | spring | 2019-04-02 | 2019-04-16 | 1 | 2 | 0 |
| | summer | 2019-07-12 | 2019-07-25 | 3 | 3 | 4 |
| | fall | 2019-11-05 | 2019-11-17 | 2 | 1 | 4 |
| 2020 | winter | 2020-01-05 | 2020-01-19 | 3 | 5 | 0 |

Sampling effort dates and on-effort, on-transect sightings for each species of interest. Winter 2019 does not appear in the table because no sightings were recorded.

seasonal average). Estimates greater than 1 indicate a positive association, and estimates less than 1 indicate a negative association.

## Model predictions

The selected amplicons summarized in the previous section and enumerated fully in the supporting information (S3 Table) were predictive of deviations of whale sightings from seasonal averages (Table 5). Combining predicted deviations with estimated seasonal trends, out-of-sample predictions of density were within 0.69–1.41 individuals per 1000 $km^2$ of observed values on average, depending on whale species and marker. For context, this result represents a 11–52% reduction in prediction error compared with imputing the seasonal average, and a 20–65% reduction in prediction error compared with carrying forward the last observation from the same season, again depending on whale species and marker. While all markers produced comparable gains in predictive power, the selected 18S-V9 amplicons yielded the best predictions for all target species.

Fig 3 shows leave-one-out predictions from all nine models compared with observed values. In particular, our models predicted spikes in density and low density with comparable accuracy. Predictions were made with comparable precision – as measured by 90% bootstrap percentile intervals – on the log scale, which translates to greater uncertainty associated with predictions of higher sightings.

 

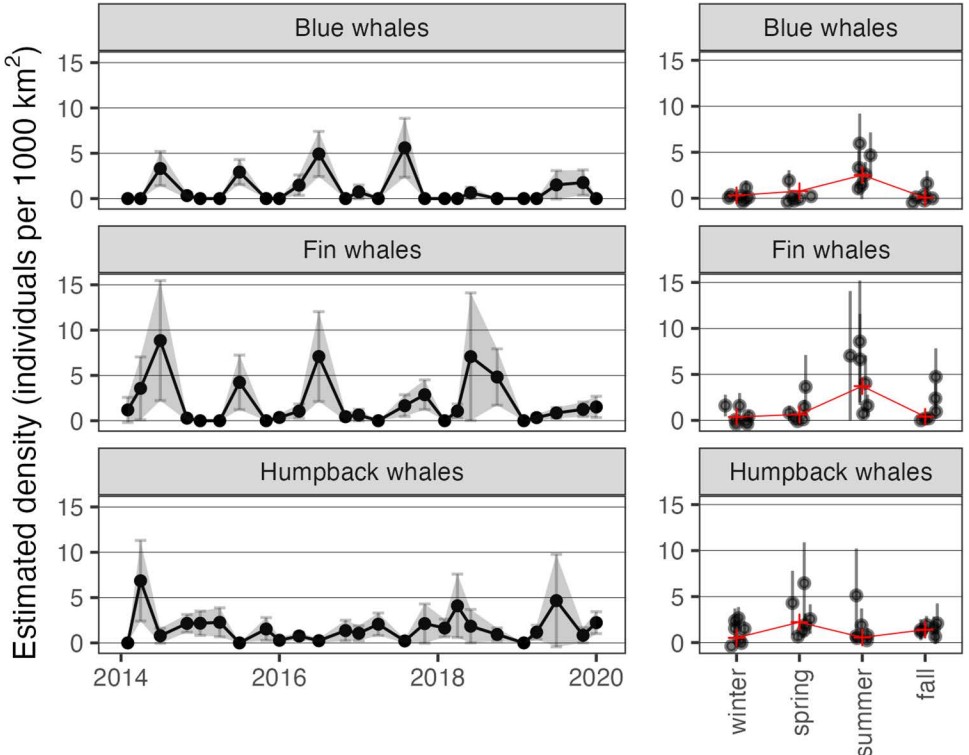

**Fig 2. Estimated density of baleen whales over time.** Estimated density (number of individuals per 1000 km$^2$) over time across years (left) and by season (right) for each whale species; seasonal averages are shown in red.

**Table 3. Fit summaries of all nine models selected in the analysis.**

| Whale species | Marker | K | Λ | Adjusted R$^2$ (log scale) | Adjusted R$^2$ (original scale) | ASVs | Families | Orders | Classes |
|---|---|---|---|---|---|---|---|---|---|
| Blue | 16S | 4 | [0.56, 0.68] | 0.955895 | 0.817874 | 48 | 28 | 21 | 12 |
| Fin | 16S | 5 | [0.56, 0.86] | 0.990324 | 0.975830 | 35 | 24 | 18 | 12 |
| Humpback | 16S | 11 | [0.62, 0.91] | 0.998393 | 0.996425 | 42 | 25 | 21 | 10 |
| Blue | 18SV4 | 10 | [0.7, 0.83] | 0.995825 | 0.927624 | 36 | 16 | 14 | 13 |
| Fin | 18SV4 | 11 | [0.65, 0.88] | 0.999514 | 0.994615 | 58 | 21 | 16 | 12 |
| Humpback | 18SV4 | 6 | [0.68, 0.74] | 0.952746 | 0.941768 | 23 | 9 | 8 | 7 |
| Blue | 18SV9 | 10 | [0.72, 0.78] | 0.995587 | 0.908874 | 23 | 12 | 11 | 10 |
| Fin | 18SV9 | 6 | [0.59, 0.72] | 0.994075 | 0.968935 | 60 | 26 | 25 | 19 |
| Humpback | 18SV9 | 7 | [0.7, 0.76] | 0.975217 | 0.966327 | 26 | 15 | 14 | 12 |

Columns show hyperparameter values ($K$, $\Lambda$), adjusted $R^2$ on both log scale and original scale along with taxonomic information for selected variables (number of ASVs selected and unique numbers of families, orders, and classes among selected ASVs).

## Communities of taxonomic annotations

A total of 148 unique taxonomic annotations were identified as predictive of baleen whales. 20% of annotations (29) were shared across all three species (blue, humpback, and fin whales), and 21% (31) of annotations were shared by two species (either blue/fin, blue/humpback, fin/humpback). Additionally, the rest (59%) were unique to a single species – 27

**Table 4. Measures of selection consistency at the ASV, family, class, and order level.**

| Whale species | Marker | J-index (ASV) | N (ASV) | J-index (Family) | N (Family) | J-index (Class) | N (Class) | J-index (Order) | N (Order) |
|---|---|---|---|---|---|---|---|---|---|
| Blue | 16S | 0.353 | 331 | 0.518 | 56 | 0.682 | 22 | 0.604 | 48 |
| Fin | 16S | 0.384 | 172 | 0.488 | 41 | 0.684 | 19 | 0.514 | 35 |
| Humpback | 16S | 0.319 | 119 | 0.229 | 35 | 0.267 | 15 | 0.258 | 31 |
| Blue | 18SV4 | 0.405 | 163 | 0.563 | 32 | 0.571 | 21 | 0.556 | 27 |
| Fin | 18SV4 | 0.405 | 232 | 0.548 | 42 | 0.485 | 27 | 0.500 | 36 |
| Humpback | 18SV4 | 0.354 | 291 | 0.457 | 46 | 0.433 | 30 | 0.447 | 38 |
| Blue | 18SV9 | 0.390 | 413 | 0.558 | 86 | 0.579 | 57 | 0.587 | 75 |
| Fin | 18SV9 | 0.352 | 145 | 0.350 | 40 | 0.370 | 27 | 0.351 | 37 |
| Humpback | 18SV9 | 0.300 | 110 | 0.517 | 29 | 0.500 | 20 | 0.500 | 26 |

For each taxonomic level, N gives the total number of unique taxa selected, and J is the proportion of those that were frequently selected across validation partitions. Hyperparameter settings were identical to those used in each of the nine models fit in the analysis.

**Table 5. Summaries of predictive performance of each model selected in the analysis.**

| Whale species | Marker | Correlation (log scale) | RMSPE (log scale) | Correlation (original scale) | RMSPE (original scale) | % reduction (lag) | % reduction (mean) |
|---|---|---|---|---|---|---|---|
| blue | 16S | 0.675 | 1.246 | 0.734 | 1.078 | 20.7 | 11.5 |
| fin | 16S | 0.939 | 0.777 | 0.907 | 1.096 | 64.7 | 52.1 |
| humpback | 16S | 0.774 | 0.838 | 0.663 | 1.238 | 42.0 | 25.8 |
| blue | 18SV4 | 0.876 | 0.839 | 0.795 | 0.990 | 27.2 | 18.7 |
| fin | 18SV4 | 0.851 | 1.025 | 0.830 | 1.415 | 54.4 | 38.2 |
| humpback | 18SV4 | 0.810 | 0.765 | 0.672 | 1.203 | 43.6 | 27.9 |
| blue | 18SV9 | 0.855 | 0.885 | 0.917 | 0.698 | 48.6 | 42.7 |
| fin | 18SV9 | 0.866 | 0.986 | 0.902 | 1.101 | 64.5 | 51.9 |
| humpback | 18SV9 | 0.876 | 0.655 | 0.746 | 1.078 | 49.4 | 35.4 |

Correlations and root mean square prediction error (RMSPE) are reported on both the logratio scale and the original scale. Percent reductions report the relative reduction in prediction error, on the original scale, of each model compared with carry-one-forward predictions (lag method) and seasonal averages (mean method).

annotations were unique to blue whales, 43 annotations were unique to fin whales, and 18 annotations were unique to humpback whales (S4 Table).

## Narrative review findings

We found 18 publications that documented 457 microbial and plankton taxa associated with blue, fin, and humpback whales, based on studies conducted in various regions throughout the world (S4 Table). After accounting for duplicate taxa, 403 unique taxa remained. These studies illustrate taxa that are known to be associated, either internally or externally, with fin, humpback, and blue whales. They highlight various aspects of baleen whale biology and ecology, including related to fecal, digestive, respiratory, and skin microbiomes; prey composition; and feeding habitats. They also document a range of epibiotic parasitic and commensal organisms, such as barnacles and diatoms. A complete list of taxa known to be associated with whales from the narrative review is contained in S4 Table.

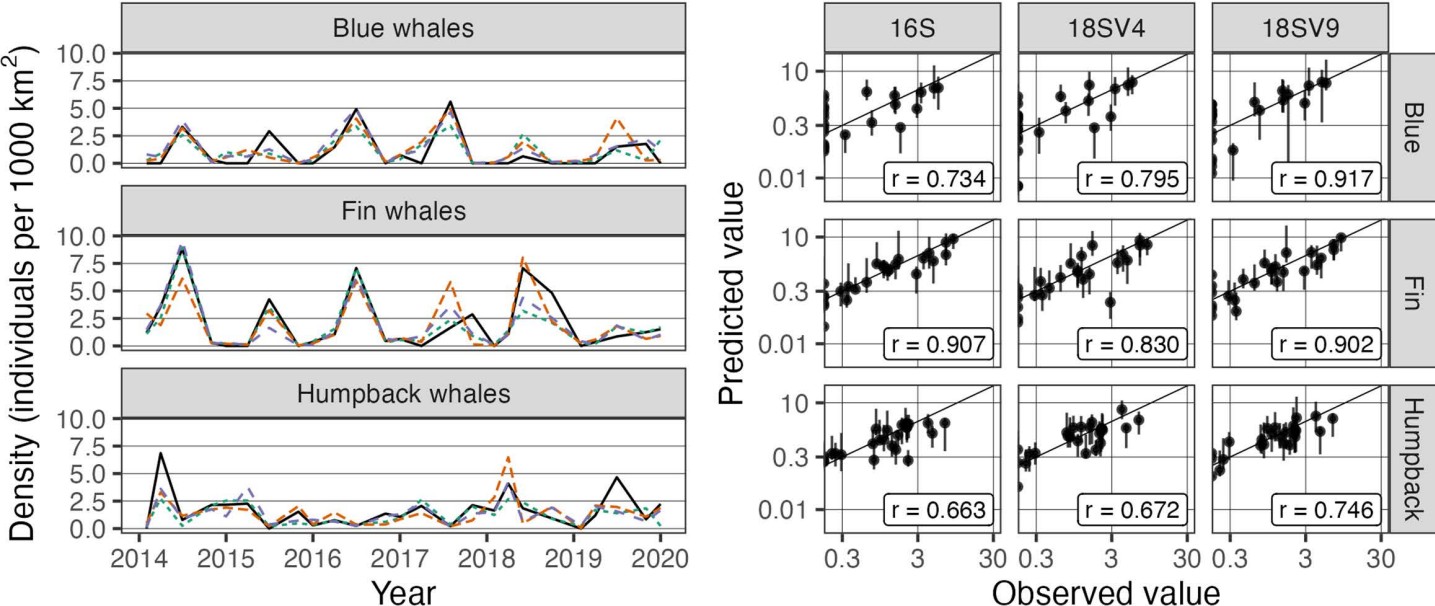

**Fig 3. Predicted density for each whale species and marker.** At left, separate time series are shown for each marker and distinguished by color and line dash. At right, separate panels compare the observations and predictions for each species/marker combination on the log scale; the solid line represents perfect predictive accuracy. The vertical line ranges show 90% bootstrap percentile intervals quantifying prediction uncertainty.

## Overlap of annotations with literature identified in narrative review

Of the 148 unique taxonomic annotations that comprised the microbial communities predictive of baleen whales in our study, 23% of the annotations (34 out of 148) were found within the existing literature exploring baleen whale microbial parasites, commensals, prey, or respiratory-associated microbes, matching at the genus (17 out of 148) or family level (17 out of 148) (S4 Table). The rest of the annotations either matched at higher taxonomic levels, including order, class, and phylum (41%; 61 out of 148) or did not have any associated matches in the literature (36%; 53 out of 148) (S4 Table).

## Discussion

In this work we leveraged the microbial and small plankton community composition for prediction of blue, fin, and humpback whale density. We found that biological communities, as identified from marker genes capturing both prokaryotic and eukaryotic plankton, were strong predictors of whale density across multiple seasons and years. To our knowledge, these are some of the first data showing that the ecological habitat of baleen whales can predict their density and track interannual variability. These results align with a growing body of evidence suggesting that baleen whale distributions in the California Current are tightly coupled to environmental conditions that drive prey availability [19,22,79], which in turn are linked to microbial and planktonic community composition.

From among six to eight thousand candidate microbial taxa per genetic marker (S2 Table), our analyses found small groups of taxa that represent the ecological habitat of baleen whales and are strongly associated with and strongly predictive of the density of blue, fin, and humpback whales (Tables 3, 5, and S3 Table). In total, these groups of ASVs

represented 148 unique taxonomic annotations identified across marker genes, with 20% shared among all three whale species, 21% shared between two species, and 59% unique to a single species. These results suggest that there is some overlap in the ecological habitat among blue, humpback, and fin whales, but each species is also related to a distinct planktonic community, with blue whales exhibiting the highest uniqueness in associated taxa.

Additionally, we compared the predictive microbial and small plankton annotations from our study to taxa found in our literature review of previously documented connections between microbes and small plankton with baleen whales. We found that 23% of the predictive taxa matched taxa from the literature we reviewed, at the genus or family level. Many of the taxa that matched are known to be prey, parasites, and commensals of baleen whales. For example, members of the genera *Sphingomonas*, *Pseudoalteromonas*, *Acinetobacter*, and *Pseudomonas* have been found in blow samples of humpback whales [33]. Additionally, members of the genus *Psychrobacter* have been found on skin and in blow samples of humpback whales [30,33] and in blow samples of blue whales [34]. *Serratia* spp. have been observed in the digestive tract of fin whales [80]. Additionally, both calanoid and cyclopoid (including Poecilostomatoida) copepods have been detected as prey in humpback whale feces [32]. In addition to these examples, many other taxa in our communities were found in the existing literature on microbial whale associates (S4 Table). Our findings suggest that some of these signals may be ecologically relevant – either directly through food-web or symbiotic interactions, or indirectly by reflecting environmental features of whale habitats, such as distinct water masses or seasonal patterns (S4 Table). Given that the rest either matched at higher taxonomic levels (41%) or had no documented matches in our literature search (36%), and some were uncultured or unassigned, this suggests that future work could examine these specific taxa further. This could include investigating some of the taxa identified in this study in greater detail to identify ecological relationships with baleen whales, applying other genetic methods to improve taxonomic resolution, or further exploring novel or understudied microbial associations relevant to whale habitats.

Ultimately, our focus is not on individual ASVs in isolation, but rather on their combined presence as a collective community. This suite of microbes and small plankton likely represents an integral component of the ecological habitat surrounding baleen whales, contributing to the complex collection of organisms that may influence or reflect their density. Our results point to a consistent community-level association between large cetaceans and microbes or small plankton, which may reflect physical and chemical signatures of oceanic water masses [81–84]. Marine microbial diversity is vast, and while marker gene amplicons (*e.g.,* 16S or 18S rRNA) are valuable for broad community characterization, they often provide insufficient resolution to infer fine-scale taxonomy or ecological niche [85–87]; thus, we refrain from presenting individual microbial taxa as predictors of whales. However, microbial communities are well-known to transform nutrients and other small molecules in ways that shape the surrounding food web [88–90]. We therefore postulate that microbial communities exist that are characteristic of habitats favorable to whale species, whether due to indirect relationships (*e.g.*, because it reflects the biochemical habitat of prey items like krill and fish) or some direct relationships (*e.g.*, microbial associates of whales).

Notably, our molecular data encompass both prokaryotic and eukaryotic plankton, which allowed us to characterize microbial diversity across different domains of life. There are likely many bacterial-bacterial and bacterial-plankton interactions, as well as other as-yet uncharacterized ecological relationships among the predictor taxa. Taken together, the community-level synergy is likely greater than the sum of its parts and encompass the holistic ecological habitat of baleen whales. We present a first step at understanding and untangling the complex linkages between the environment, base of the food web, and top consumers. However, further research is needed to more deeply examine the groups of ASVs that collectively define these microbial communities.

Our statistical analysis framework combines log-ratio methodology for compositional data analysis [71,72] with sparse partial least squares [78] for interpretable dimension reduction, and we used stability selection [77] to improve the robustness of the variable selection procedure to small perturbations in training data. We found that the resulting estimation procedure in our analysis exhibits a range of selection consistencies, depending on the model. Among the 9 models in our analysis, anywhere from an estimated 30–40% of ASVs were consistently selected when holding out one data point

at a time (Table 3). Although surprisingly low, given that stability methods aim precisely at achieving consistent selection, there are several possible explanations for this result. First, the modest sample size entails that each data point exerts considerable influence on model fit. Among the 25 cruises in our analysis, one observation constitutes 4% of available data. Second, the density estimates are sparse – most cruises record few sightings and the limited number of cruises per year (~4 per year) may mean that we miss the full range of whale densities in a given year (e.g., minimum or maximum). Although spikes in sightings comprise only 8–16% of available data, depending on whale species, it is reasonable to speculate that these cruises capture the most information about potential ecological correlations. By comparison with the remaining 84–92% of the data, removal of one or two of these high-sightings observations substantially alters the variation in the time series. Thus, depending on which observations are held out, fitted models may describe fundamentally different ecological processes—either small variations among low density or large fluctuations between high and low densities. Third, our analysis did not account for (a) uncertainty in taxonomic classification of ASVs or (b) potential biases inherent to eDNA methods. These include extraction, primer, and amplification biases, which can cause some species to be missed and others to appear more or less abundant than they really are. Strong correlations among amplicon relative abundances due to either factor could lead to instability in variable selection under small perturbations of the data; this is a well-documented phenomenon in the statistical literature [77,91,92]. Fourth, partial least squares estimation is known to be sensitive to outliers—a fact which has produced proposals for robust PLS estimators [93]. It is therefore plausible that sparse partial least squares would exhibit high selection variability under data perturbations, especially in light of the data sparsity discussed above. Finally, varying uncertainty in density estimation by cruise (Fig 2) is not accounted for in the modeling framework, but may produce uneven signal strength of associations between microbial communities and baleen whale density depending on which cruises are used to fit models. All of these factors may contribute to the wide range of selection consistency observed in our work.

Overall, our findings suggest that planktonic communities can serve as a predictor of baleen whale densities. This study expands on previous research focused on specific planktonic prey as whale predictors to integrate the full planktonic community, including direct and indirect relationships to baleen whales, as an ecological habitat predictor. This study presents a reliable approach to predict baleen whale densities across long temporal (6 years) and spatial scales (over 200,000km$^2$) using metabarcoding-derived communities of microbes and small plankton found in the water column.

By demonstrating links between microbial community composition and large whale density, this work can be used to generate potential explanatory variables to predict cetacean density. Current habitat and density surface models typically explain only a small proportion of variation in the input data, which is often derived from line transect surveys. Given the low detectability of whales and the challenges of sampling their eDNA, plankton eDNA may serve as a complementary proxy for predicting whale density. Thus, planktonic community measures may help to explain more of the variation in whale density and help to improve existing density surface models (e.g., [66]). For example, a microbial community index could be used to characterize the ecological habitat of specific whale species. Establishing such an index offers an additional tool for monitoring and managing whale populations.

This work can also inform hypotheses about the ecological relationships between whales and bacterioplankton, phytoplankton, and zooplankton. Additionally, important insights can be gained about planktonic communities that support higher trophic levels off the coast of California, effectively serving as ecological "fingerprints" of habitat suitability and to monitor changes in habitat quality. This also highlights the potential for using whales and their ecological habitats as sentinels for detecting and tracking changes in marine ecosystems.

This holistic predictive framework is broadly transferable beyond marine mammal research, management, and conservation. For example, future work could leverage large microbiome programs like Tara Oceans Expedition or Earth Microbiome Project to identify ecological habitats relevant to the conservation, population management, or reintroduction efforts of other species. We believe that assessing the overall ecological habitat offers a holistic predictive approach that can be extended and tested for other species, ecosystems, regions, and over longer time scales.

## Supporting information

**S1 Table. Numbers of NCOG samples tabulated by marker, cruise, and transect.**
(XLSX)

**S2 Table. Taxonomic annotations of candidate ASVs used in the analysis.** Table divided into (a) 16S, (b) 18S-V4, and (c) 18S-V9 markers.
(XLSX)

**S3 Table. Estimated model coefficients and taxonomic annotations of selected ASVs used for prediction.** Table divided into (a) 16S, (b) 18S-V4, and (c) 18S-V9 models.
(XLSX)

**S4 Table. Summary of narrative review findings.** Table divided into (a) taxa identified by our models with supporting citations from the existing literature, if any, and (b) taxa reported in the existing literature with known direct relationships between blue, humpback, and fin whales and bacteria, microbes, and small plankton.
(XLSX)

**S5 Table. Summary of density estimation results.** (a) per-species sample size of detections before and after truncation for line transect analysis; (b) list of all line transect detection function models fitted for each species, with corresponding AIC values and delta AIC (*i.e.*, the difference in AIC between that model and the lowest-AIC model for that species).
(XLSX)

**S1 Fig. Residual diagnostic checks for density models.** A) residuals against fitted values for each model (whale species and marker combination) and B) partial autocorrelations for each model to assess possible time dependence across successive surveys not accounted for in the models.
(TIFF)

## Acknowledgments

We are thankful to Matthew Robbins for his support in developing code to programmatically retrieve taxonomic information from the World Register of Marine Species (WoRMS) about taxa in our literature review. We are grateful to Ryan Kelly, Ole Shelton, and Eiren Jacobson for their valuable input. We also thank the reviewers (Matthew Harke, Chloe Robinson, and an anonymous reviewer) and the editor for their constructive comments and suggestions that improved the manuscript. We are grateful to all of the people who helped to make the CalCOFI cruises and associated sample and data collection possible. We are thankful to John Hildebrand and Joshua Jones for the continued support in organizing marine mammal visual observations during CalCOFI cruises, and Katherine Whitaker as well as other observers who collect the visual observations at sea.

## Author contributions

**Conceptualization:** Erin V. Satterthwaite, Nastassia V. Patin, Michaela N. Alksne, Julie Dinasquet, Andrew E. Allen, Brice X. Semmens.

**Data curation:** Robert H. Lampe.

**Formal analysis:** Trevor D. Ruiz, Len Thomas, Katherine G. Chan, Nicholas A. Patrick.

**Investigation:** Robert H. Lampe, Andrew E. Allen, Simone Baumann-Pickering.

**Methodology:** Erin V. Satterthwaite, Trevor D. Ruiz, Michaela N. Alksne, Len Thomas, Katherine G. Chan, Nicholas A. Patrick.

**Project administration:** Erin V. Satterthwaite, Trevor D. Ruiz.

**Software:** Trevor D. Ruiz, Len Thomas, Katherine G. Chan, Nicholas A. Patrick.

**Validation:** Trevor D. Ruiz.

**Visualization:** Trevor D. Ruiz, Nicholas A. Patrick.

**Writing – original draft:** Erin V. Satterthwaite, Trevor D. Ruiz, Nastassia V. Patin, Michaela N. Alksne, Len Thomas, Robert H. Lampe.

**Writing – review & editing:** Erin V. Satterthwaite, Trevor D. Ruiz, Nastassia V. Patin, Michaela N. Alksne, Len Thomas, Julie Dinasquet, Robert H. Lampe, Simone Baumann-Pickering, Brice X. Semmens.

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
