## [Decision Letter · Decision Letter 0]

30 Oct 2025

Dear Dr. Satterthwaite,

Thank you for submitting your manuscript to PLOS ONE. After careful consideration, we feel that it has merit but does not fully meet PLOS ONE’s publication criteria as it currently stands. Therefore, we invite you to submit a revised version of the manuscript that addresses the points raised during the review process.

We look forward to receiving your revised manuscript.

Kind regards,

Vitor Hugo Rodrigues Paiva, Ph.D.

Academic Editor

PLOS ONE

Journal Requirements:

“This material is based upon research supported by the Office of Naval Research under Award Number (N00014-22-1-2719), upon work supported by National Oceanic and Atmospheric Administration grants NA15OAR4320071 and NA19NOS4780181 and Simons Foundation Collaboration on Principles of Microbial Ecosystems (PriME) grant 970820. EVS was supported by a partnership among CalCOFI participants, including Scripps Institution of Oceanography (SIO), NOAA Southwest Fisheries Science Center (NA20OAR4170258), California Department of Fish and Wildlife (#P2370002), and California Sea Grant (NA22OAR4170106).”

Reviewer's Responses to Questions

**Comments to the Author**

1. Is the manuscript technically sound, and do the data support the conclusions?

Reviewer #1: Yes

Reviewer #2: Yes

2. Has the statistical analysis been performed appropriately and rigorously?

Reviewer #1: Yes

Reviewer #2: Yes

3. Have the authors made all data underlying the findings in their manuscript fully available?

Reviewer #1: Yes

Reviewer #2: Yes

4. Is the manuscript presented in an intelligible fashion and written in standard English?

Reviewer #1: Yes

Reviewer #2: Yes

Reviewer #1: The study uses a dataset from the CalCOFI monitoring program to model relative abundances of bacteria and small plankton inferred from metabarcoding with baleen whale observations (here blue, fin, and humpback) to understand if assessing microbial and small plankton communities can be used for predicting baleen whale densities. Overall, the study is robust, and I only had a few questions on the methods and how some choices might influence the overall model results. A very interesting approach with potential application to other systems (e.g. could we use this for fish? Other marine mammals?), but I wonder how well it will perform under longer time scales or other regions. The authors do a fantastic job of discussing the challenges of this approach and the need for further research to validate their findings.

Line 30 – forgive my ignorance, but what is RMSE? Consider defining here as you have done with ASV on line 26

Line 158 –curious if 56 deg C was used for the 16S? the Parada set typically uses an annealing temp closer to 50 deg C and wonder how this might impact observed biodiversity and subsequent patterns and results.

182 – In considering the method used to count cetaceans, I completely understand the reasoning for using passing mode (as explained in Campbell et al.) but also note that they acknowledge that it results in “more unidentified or mis-identified groups, more biased estimates of group size, and less precise species percentages than closing mode”. How might this impact marine mammal density estimates and subsequently the results presented? The authors do a fantastic job of accounting for other biases introduced by the data (e.g. seasonal influence, metabarcoding) in their model but I wonder if this is accounted for as well. Certainly counting marine mammals at sea is difficult, so I appreciate the level of scrutiny the authors impose, just more curious about this than anything.

Reviewer #2: This study presents an innovative and rigorous approach to predicting baleen whale population densities using microbial and small plankton communities as ecological indicators. The concept of assessing microbial assemblages (via eDNA metabarcoding) to infer the distribution and/or abundance of higher trophic species (cetaceans, elasmobranchs, etc) is highly relevant and aligns with current advances in molecular ecology and ecosystem modeling. Overall, the paper offers an impressive integration of visual sightings data metabarcoding, and statistical modeling, and it makes a valuable contribution to marine eDNA research as well as cetacean ecology.

However, several sections of the manuscript would benefit from further clarification and reorganization to strengthen its overall coherence, especially for a broad journal audience like that of PLOS. The Introduction section requires more context and flow to clearly establish the study rationale, objectives, and novelty. Concepts that are critical to the study, such as eDNA, metabarcoding, and the ecological framing of “microbial habitats”, should be introduced earlier and defined clearly to guide readers who may be unfamiliar with these concepts. The Methods section, while generally sound, needs substantially more detail on laboratory procedures, particularly regarding contamination avoidance measures, sample handling, and sequencing protocols, to ensure transparency. The statistical analyses are a clear strength of the paper. The modeling framework is sophisticated and very well-explained. The results are also presented nicely and highlight meaningful ecological relationships across trophic levels.

The Discussion, while well-written, could go further in connecting the findings to broader ecological and conservation implications. Including a “future outlook” section that considers the applications of this predictive framework, for instance, in whale population monitoring and management, would enhance the manuscript’s impact.

In summary, this is a well-designed study that contributes greatly to the understanding of whale-microbe ecological linkages. With a more cohesive introduction, expanded methodological detail, and a stronger framing of broader applications, the manuscript will be suitable for publication. Please also address my line-by-line feedback attached to this review.

.

Reviewer #1: **Yes:**Matthew HarkeMatthew HarkeMatthew HarkeMatthew Harke

Reviewer #2: No

---

## [Author Response · Author response to Decision Letter 1]

9 Jan 2026

Response to reviewers

PONE-D-25-51982

We would like to thank you and the reviewers for their thoughtful review, which has substantially improved our manuscript. We appreciate the time and effort you took to review our work, and we have carefully addressed all the comments and suggestions. Our responses are provided in italicized font below and highlighted via track changes in the attached document. All references to lines below refer to the new document line numbers (with track changes).

Thank you!

Journal Requirements

We have updated the manuscript formatting following the provided guides and implemented the following changes:

- Remove numbering and adjust Level 1 and Level 2 headers to sentence case in 18pt and 16pt font size, respectively

- Format figure and table titles in bold and add legends for figures and tables appearing in the body of the paper

- Change figure citations from Figure 1, Figure 2, etc., to Fig 1, Fig 2, etc.

- Change supplementary figure/table citations to S1 Fig, S1 Table, etc.

- Update author byline symbol for equal contributions and corresponding author

- Add corresponding author email

- Add departments to author affiliations where missing

We are holding on updating author-year citations to numbered citations pending any further revisions requested by reviewers that may alter the number and order of appearance of references.

“This material is based upon research supported by the Office of Naval Research under Award Number (N00014-22-1-2719), upon work supported by National Oceanic and Atmospheric Administration grants NA15OAR4320071 and NA19NOS4780181 and Simons Foundation Collaboration on Principles of Microbial Ecosystems (PriME) grant 970820. EVS was supported by a partnership among CalCOFI participants, including Scripps Institution of Oceanography (SIO), NOAA Southwest Fisheries Science Center (NA20OAR4170258), California Department of Fish and Wildlife (#P2370002), and California Sea Grant (NA22OAR4170106).”

We have included the suggested statement regarding funders’ role: "The funders did not have a role in study design, data collection and analysis, decision to publish, or preparation of the manuscript." We also would like to add “This research is partly based upon work supported by the Research, Scholarly & Creative Activities Program awarded by the Cal Poly Division of Research.”

The complete funding statement should read “This material is based upon research supported by the Office of Naval Research under Award Number (N00014-22-1-2719), upon work supported by National Oceanic and Atmospheric Administration grants NA15OAR4320071 and NA19NOS4780181 and Simons Foundation Collaboration on Principles of Microbial Ecosystems (PriME) grant 970820. EVS was supported by a partnership among CalCOFI participants, including Scripps Institution of Oceanography (SIO), NOAA Southwest Fisheries Science Center (NA20OAR4170258), California Department of Fish and Wildlife (#P2370002), and California Sea Grant (NA22OAR4170106). This research is partly based upon work supported by the Research, Scholarly & Creative Activities Program awarded by the Cal Poly Division of Research. The funders did not have a role in study design, data collection and analysis, decision to publish, or preparation of the manuscript.”

Data are currently publicly available at https://doi.org/10.5281/zenodo.15678927; we have amended the data availability statement accordingly.

No reviewer comments included a recommendation to cite specific previously published works.

Reviewer #1

The study uses a dataset from the CalCOFI monitoring program to model relative abundances of bacteria and small plankton inferred from metabarcoding with baleen whale observations (here blue, fin, and humpback) to understand if assessing microbial and small plankton communities can be used for predicting baleen whale densities. Overall, the study is robust, and I only had a few questions on the methods and how some choices might influence the overall model results. A very interesting approach with potential application to other systems (e.g. could we use this for fish? Other marine mammals?), but I wonder how well it will perform under longer time scales or other regions. The authors do a fantastic job of discussing the challenges of this approach and the need for further research to validate their findings.

Thank you. We have integrated the idea expressed in the comment: “A very interesting approach with potential application to other systems (e.g. could we use this for fish? Other marine mammals?), but I wonder how well it will perform under longer time scales or other regions” into the concluding sentence (lines 1108-1113).

Line 30 – forgive my ignorance, but what is RMSE? Consider defining here as you have done with ASV on line 26

We have replaced the acronym with “root mean square prediction error”.

Line 158 –curious if 56 deg C was used for the 16S? the Parada set typically uses an annealing temp closer to 50 deg C and wonder how this might impact observed biodiversity and subsequent patterns and results.

The annealing temperature depends on MgCl2 concentrations in PCR buffer which differ between our study and that used by Parada et al. (2016). In accordance with the higher salt concentration in our PCR buffer, our annealing temperature is higher. The master mix used by Parada et al. (2016) recommends a minimum annealing temperature of 50°C, whereas ours recommends a minimum of 57°C; thus, we believe that our annealing temperature of 56°C is appropriately similar, while being high enough to minimize off-target amplification.

182 – In considering the method used to count cetaceans, I completely understand the reasoning for using passing mode (as explained in Campbell et al.) but also note that they acknowledge that it results in “more unidentified or mis-identified groups, more biased estimates of group size, and less precise species percentages than closing mode”. How might this impact marine mammal density estimates and subsequently the results presented? The authors do a fantastic job of accounting for other biases introduced by the data (e.g. seasonal influence, metabarcoding) in their model but I wonder if this is accounted for as well. Certainly counting marine mammals at sea is difficult, so I appreciate the level of scrutiny the authors impose, just more curious about this than anything.

Because we did not correct for the percentage of unidentified animals among those sighted in the surveys, our estimates likely underestimate the true density. However, the relative seasonal estimates and contributions of each species wouldn't change that much if unidentified detection rates are similar across species.

Reviewer #2

This study presents an innovative and rigorous approach to predicting baleen whale population densities using microbial and small plankton communities as ecological indicators. The concept of assessing microbial assemblages (via eDNA metabarcoding) to infer the distribution and/or abundance of higher trophic species (cetaceans, elasmobranchs, etc) is highly relevant and aligns with current advances in molecular ecology and ecosystem modeling. Overall, the paper offers an impressive integration of visual sightings data metabarcoding, and statistical modeling, and it makes a valuable contribution to marine eDNA research as well as cetacean ecology.

However, several sections of the manuscript would benefit from further clarification and reorganization to strengthen its overall coherence, especially for a broad journal audience like that of PLOS.

The Introduction section requires more context and flow to clearly establish the study rationale, objectives, and novelty. Concepts that are critical to the study, such as eDNA, metabarcoding, and the ecological framing of “microbial habitats”, should be introduced earlier and defined clearly to guide readers who may be unfamiliar with these concepts.

We have substantially revised the Introduction to establish the study rationale and objectives, clarify the novelty, and introduce key concepts (e.g., eDNA/metabarcoding, and the ecological habitat concept) (lines 242-252; 282-313; and 426-463).

The Methods section, while generally sound, needs substantially more detail on laboratory procedures, particularly regarding contamination avoidance measures, sample handling, and sequencing protocols, to ensure transparency.

We have substantially revised the Methods section to include more detail on laboratory procedures (lines 490-520), particularly regarding contamination avoidance measures (lines 495-496), sample handling (lines 490-493), and sequencing protocols (lines 514-520).

The statistical analyses are a clear strength of the paper. The modeling framework is sophisticated and very well-explained. The results are also presented nicely and highlight meaningful ecological relationships across trophic levels.

The Discussion, while well-written, could go further in connecting the findings to broader ecological and conservation implications. Including a “future outlook” section that considers the applications of this predictive framework, for instance, in whale population monitoring and management, would enhance the manuscript’s impact.

We have substantially revised the text in the Discussion to connect the findings to broader applications and implications of the work. Please see lines 1072 – 1113.

In summary, this is a well-designed study that contributes greatly to the understanding of whale-microbe ecological linkages. With a more cohesive introduction, expanded methodological detail, and a stronger framing of broader applications, the manuscript will be suitable for publication. Please also address my line-by-line feedback attached to this review.

Thank you. We have addressed the broader comments related to the introduction, expanded methodological detail, and a stronger framing of broader applications above and in the line by line comments below.

ABSTRACT

● Line 14: Would you be able to change the wording in the first sentence? As it is worded, “density of marine mammals” sounds like the framework for a physiological paper.

We changed to distribution and abundance.

● Lines 15-18: Consider removing the parenthetical phrase “– the potential ‘ecological habitat’ –” to improve sentence clarity and flow. It feels redundant given the context and interrupts the main idea.

Removed.

● Line 20: can you please elaborate on “each quarterly season” or change the wording?

We have changed this to “each season”.

● Line 21: Which densities were estimated?

We have clarified by adding “Densities of Balaenopteridae whales”.

● Lines 24-25: please include the words “environmental DNA” in this sentence.

We have included “environmental DNA” in the sentence to read “We identified microbial and small plankton communities from environmental DNA that were specific to each target species and were strong predictors of estimated density”.

● Line 26: You do not need to provide an acronym for ASVs if it is not used again in the abstract (saves word count).

We have removed the acronym as suggested.

● Lines 28-30: what do you mean by “naive seasonal carry-forward and seasonal averaging approaches to prediction”? I’m not sure that this is the correct phrasing… please change or remove this.

We have revised this sentence to improve clarity, replacing it with: “Our approach improves out-of-sample root mean square prediction error by up to 65% compared with simple alternative methods.”

● I suggest you check word count for the journal abstract, this seems a bit long and it’s generally unusual to have two paragraphs in an abstract.

Thank you for catching this. We have shortened the abstract to less than 300 words (as per guidance by the PLOS One submission guidelines).

● You may want to add “metabarcoding” to the keywords and remove “quantitative

Ecology”.

We have removed “quantitative ecology” and added “metabarcoding” to the keywords.

INTRODUCTION

● Line 52: Provide cultural value to whom or what?

We added “to people” since aesthetic, spiritual, educational, and recreational experiences contribute to their cultural value for human societies.

● The first paragraph should flow a bit more cohesively. This can be achieved by including a sentence that connects the conservation/management idea with the three study species and then another sentence which connects the study species to your study area. As it is written, the California Current is suddenly introduced without a purpose.

We have substantially edited the Introduction so that it flows from the conservation management idea to the three species and study area. Please see lines 242 – 252; 253-284; 426 – 428.

● Line 63: I suggest not using the word “track”, as the whales are not directly “tracking” the krill species. Perhaps instead use “and are correlated with present aggregations”.

We edited to “their distributions tend to be correlated with krill aggregations” since they are most likely following the krill or something that is very closely associated with them (likely an acoustic or chemical signature).

● Beware of introducing new species throughout your manuscript – there is currently an unsystematic pattern of using Latin vs common names.

We follow the general convention of using the full common name followed by the scientific name at first mention (e.g., “Blue whale (Balaenoptera musculus)”). After first mention, we use common names (e.g., “blue whales”). The only exception to this convention involves taxa that do not have a well-known common name like Thysanoessa spinifera. For these species, we provide only the scientific names.

● Line 77: Do you mean habitat-specific ecology?

Yes. We have changed “biological” to “ecological”.

● Line 78: The work is not particularly novel - there are many studies linking microbes/plankton presence and density to whale presence (e.g., Friedlaender et al., 2006; Fielder et al., 2018; Pendleton et al., 2012). You could rewrite this to indicate that the novel approach is based on metabarcoding data, but this has still been done before... (e.g., Carroll et al., 2019; Visser et al., 2021; Boyse et al., 2024). If this study is truly novel, please describe how in more detail to convince readers

We have clarified that our approach builds on these existing studies and efforts to assess the ability of the entire planktonic community to predict the den

---

## [Decision Letter · Decision Letter 1]

9 Feb 2026

Dear Dr. Satterthwaite,

Thank you for submitting your manuscript to PLOS ONE. After careful consideration, we feel that it has merit but does not fully meet PLOS ONE’s publication criteria as it currently stands. Therefore, we invite you to submit a revised version of the manuscript that addresses the points raised during the review process.

We look forward to receiving your revised manuscript.

Kind regards,

Vitor Hugo Rodrigues Paiva, Ph.D.

Academic Editor

PLOS One

Journal Requirements:

Reviewers' comments:

Reviewer's Responses to Questions

**Comments to the Author**

Reviewer #2: (No Response)

Reviewer #3: All comments have been addressed

2. Is the manuscript technically sound, and do the data support the conclusions?

Reviewer #2: Yes

Reviewer #3: Yes

3. Has the statistical analysis been performed appropriately and rigorously?

Reviewer #2: Yes

Reviewer #3: Yes

4. Have the authors made all data underlying the findings in their manuscript fully available?

Reviewer #2: Yes

Reviewer #3: Yes

5. Is the manuscript presented in an intelligible fashion and written in standard English?

Reviewer #2: Yes

Reviewer #3: Yes

Reviewer #2: I thank the authors for their thorough and thoughtful revision of their manuscript. I have only a few minor comments for consideration before the final version is published.

ABSTRACT

Line 24-28: The microbes/plankton communities are not directly predicting whale density, rather, they may influence whale behavior and distribution, with the prediction performed by the modeling approach applied in the study.

Line 32: “in” rather than “using”

Line 38: Please add a comma between “parasite” and “or” to follow your sentence structure.

INTRODUCTION

I’m sorry! I am still wary of the first sentence as it’s the first instance in which you can draw in readers and establish credibility for the study. At first, you state that larger baleen whales play a vital role in marine ecosystems, but the supporting point for this introductory statement is that they “provide cultural value to people…” which is not a role in a marine ecosystem, rather an anthropogenic effect of whales. Please change this sentence.

Lines 88-94: I think this is a better place (rather than lines 111-113) to introduce eDNA as you are introducing metabarcoding here (given that the journal audience may not know about eDNA and/or metabarcoding).

METHODS

Lines 153-157: Is it possible to state how often these work spaces were cleaned and at which percentage was the ethanol?

Line 172: Which ratio of beads:product was used?

Line 176: Okay, I see 0.8x was used, perhaps you could also state this above.

Line 184: In my initial comment here, I asked about overlapping ASVs. I had meant overlapping species (and/or genus, family) assignments (from ASVs) that were identified with both markers. For example, it would be interesting to know how many of x plankton species (or genus) were identified across each gene of interest. However, this is not critical to the outcomes of this paper and is not necessary to be included.

RESULTS

No further comments.

DISCUSSION

Line 542: “strong” would be a better word than “good” here.

Lines 549-556: It is not usual to include further numerical results and references to tables in the discussion. I see that PLOS does have flexibility for a “Results and Discussion” section, but as you have kept them separate, I am wondering if some of the information in this paragraph should be kept in the Results section rather than repeated here. I don’t think there is a “good” or “bad” way to go about this, and if you and your co-authors agree that the data should be kept in this part of the discussion, then that is okay, but I figured that I would note this. 4

Lines 590-592: This statement may read a bit strong as marker genes can still be informative at various taxonomic scales. You may want to slightly de-escalate the wording to emphasize limitations in fine-scale taxonomic/ecological resolution rather than unreliability. The sentence could be rephrased: Marine microbial diversity is vast, and while marker-gene amplicons (e.g. 16S or 18S rRNA) are valuable for broad community characterization, they often provide insufficient resolution to infer fine-scale taxonomy or ecological niche (83-85).

FIGURES

A small note regarding the north arrow and lat/long: I did not mean to come off as condescending here - as someone with a degree in GIS, north arrows were one of the “requirements” that I was trained to include in any map(s) that are generated for publication. I do agree that latitude and longitude denote coordinate location sufficiently for readers, and upon reading up on this, it seems that over the past years, north arrows are no longer seen as one of the four key elements (scale bar, north arrow, legend, coordinate grid) of maps. So thank you for the learning opportunity and I will keep this in mind as I review other papers!

Reviewer #3: General comments

From reviewing the response to the previous reviewer comments, the authors have done a good job incorporating the feedback into this version of the manuscript. I have therefore provided my review on the ‘revised manuscript with tracked changes’ version.

Overall, the manuscript is well-written, interesting, and scientifically sound. I have made a couple of very minor comments below. Congrats to the authors on such a thorough, impactful study!

Specific comments

Lines 70-71: May be best to capitalize listing status (check journal requirements however)

Lines 218-220: Is there a reason you didn’t use bleach (or similar) to fully decontaminate sampling bottles? As you didn’t collect sampling blanks, it is hard to determine if any species for sure were not carried over between the sampling events

.

Reviewer #2: No

Reviewer #3: **Yes:**Chloe V RobinsonChloe V RobinsonChloe V RobinsonChloe V Robinson

---

## [Author Response · Author response to Decision Letter 2]

3 Mar 2026

Dear Dr. Paiva,

Thank you for the invitation to revise and resubmit our manuscript. We have revised the manuscript in response to the reviewers’ comments and believe we have addressed the remaining concerns. Below, we provide a detailed response to each comment, and we attach copies of the revised manuscript with and without tracked changes.

We appreciate your continued consideration of our manuscript for publication in PLOS One.

We also note that “The funders had no role in study design, data collection and analysis, decision to publish, or preparation of the manuscript” so we request that this be added to our online submission.

Sincerely,

Dr. Erin Satterthwaite & Dr. Trevor Ruiz

Reviewer #2

I thank the authors for their thorough and thoughtful revision of their manuscript. I have only a few minor comments for consideration before the final version is published.

ABSTRACT

Line 24-28: The microbes/plankton communities are not directly predicting whale density, rather, they may influence whale behavior and distribution, with the prediction performed by the modeling approach applied in the study.

We agree that microbial and plankton communities are not directly determining whale density. However, we use “predict” in the statistical sense and we are careful to avoid causal language throughout the abstract. Here at the outset we prefer to retain “predict” rather than add qualifiers that might obscure the sentence, but have added a clarification to the fourth sentence of the abstract that we intend statistical rather than causal or direct prediction.

Line 32: “in” rather than “using”

We have adopted the suggestion.

Line 38: Please add a comma between “parasite” and “or” to follow your sentence structure.

We have adopted the suggestion.

INTRODUCTION

I’m sorry! I am still wary of the first sentence as it’s the first instance in which you can draw in readers and establish credibility for the study. At first, you state that larger baleen whales play a vital role in marine ecosystems, but the supporting point for this introductory statement is that they “provide cultural value to people…” which is not a role in a marine ecosystem, rather an anthropogenic effect of whales. Please change this sentence.

We have rearranged the sentence to make clear that we are making separate statements about the significance of whales in ecological, cultural, and conservation contexts. It now reads, “Large baleen whales play a vital role in marine ecosystems by helping to regulate ecosystem processes [1]. They are of conservation and management relevance, as any populations are listed as threatened (Vulnerable, Endangered, or Critically Endangered) on the IUCN Red List of Threatened Species [2], and they provide significant cultural value to people [3].”

Lines 88-94: I think this is a better place (rather than lines 111-113) to introduce eDNA as you are introducing metabarcoding here (given that the journal audience may not know about eDNA and/or metabarcoding).

We prefer to retain the current narrative structure, but have revised this sentence to remove the term “metabarcoding” and replaced with “molecular methods” so that it is not introduced before it is defined clearly.

METHODS

Lines 153-157: Is it possible to state how often these work spaces were cleaned and at which percentage was the ethanol?

We used 70% ethanol and cleaned the workspace before each use.

Line 172: Which ratio of beads:product was used?

We have clarified that 1x was used.

Line 176: Okay, I see 0.8x was used, perhaps you could also state this above.

As noted above we have revised as suggested.

Line 184: In my initial comment here, I asked about overlapping ASVs. I had meant overlapping species (and/or genus, family) assignments (from ASVs) that were identified with both markers. For example, it would be interesting to know how many of x plankton species (or genus) were identified across each gene of interest. However, this is not critical to the outcomes of this paper and is not necessary to be included.

Since the line reference pertains to the molecular methods rather than analysis, we feel that this information would be more appropriate to include in the Results section and have added a brief comment at the first mention of S2 Table containing taxonomic annotations. We are careful to clarify in this addition that the overlap percentages reported are among candidates used in the models, i.e., after relative abundance filtering, with the implication that the percentages reported should not be viewed as accurate estimates of overlap among all detections across markers. We feel that the post-filtering overlap is more salient to the analysis in the paper than the pre-filtering overlap. That said, however, the same summary statistics among annotations prior to filtering are 12.8%, 3.9%, and 44.7% (compared with 9.9%, 0.7%, 51.2%), so while the filtering evidently introduces some bias to the overlap measures, it is not so substantial as to alter the relative interpretation of the extent of overlap across markers.

RESULTS

No further comments.

DISCUSSION

Line 542: “strong” would be a better word than “good” here.

We have adopted the suggestion.

Lines 549-556: It is not usual to include further numerical results and references to tables in the discussion. I see that PLOS does have flexibility for a “Results and Discussion” section, but as you have kept them separate, I am wondering if some of the information in this paragraph should be kept in the Results section rather than repeated here. I don’t think there is a “good” or “bad” way to go about this, and if you and your co-authors agree that the data should be kept in this part of the discussion, then that is okay, but I figured that I would note this. 4

We appreciate the note and agree that the discussion reads more clearly without the two sentences that reiterate the quantitative findings of the paper. These have been removed.

Lines 590-592: This statement may read a bit strong as marker genes can still be informative at various taxonomic scales. You may want to slightly de-escalate the wording to emphasize limitations in fine-scale taxonomic/ecological resolution rather than unreliability. The sentence could be rephrased: Marine microbial diversity is vast, and while marker-gene amplicons (e.g. 16S or 18S rRNA) are valuable for broad community characterization, they often provide insufficient resolution to infer fine-scale taxonomy or ecological niche (83-85).

We have adopted the suggestion.

FIGURES

A small note regarding the north arrow and lat/long: I did not mean to come off as condescending here - as someone with a degree in GIS, north arrows were one of the “requirements” that I was trained to include in any map(s) that are generated for publication. I do agree that latitude and longitude denote coordinate location sufficiently for readers, and upon reading up on this, it seems that over the past years, north arrows are no longer seen as one of the four key elements (scale bar, north arrow, legend, coordinate grid) of maps. So thank you for the learning opportunity and I will keep this in mind as I review other papers!

We appreciate the clarification and were not aware that this was a strong convention among geographers or those with GIS training, so the reviewer’s comment serves as a learning opportunity for us as well.

Reviewer #3

General comments

From reviewing the response to the previous reviewer comments, the authors have done a good job incorporating the feedback into this version of the manuscript. I have therefore provided my review on the ‘revised manuscript with tracked changes’ version.

Overall, the manuscript is well-written, interesting, and scientifically sound. I have made a couple of very minor comments below. Congrats to the authors on such a thorough, impactful study!

Specific comments

Lines 70-71: May be best to capitalize listing status (check journal requirements however)

We have amended the sentence to include the exact listing categories that correspond to “threatened” parenthetically and have capitalized those categories.

Lines 218-220: Is there a reason you didn’t use bleach (or similar) to fully decontaminate sampling bottles? As you didn’t collect sampling blanks, it is hard to determine if any species for sure were not carried over between the sampling events

It was not feasible to use bleach or another cleaning agent in between each collection event. We also now clarify that the bottles were rinsed with the sample three times before sampling. The Milli-Q rinse likely not only removed cells from the wash itself, but also from osmotic shock as these are marine organisms. Further rinsing with the sample before filling provides the opportunity to remove any remaining organisms from the bottle. Even if something remained in the bottle, its abundance relative to the sample, which was multiple liters of seawater, certainly was drastically reduced.

---

## [Editor Report · Decision Letter 2]

9 Mar 2026

Microbial and small zooplankton communities predict density of baleen whales in the southern California Current Ecosystem

PONE-D-25-51982R2

Dear Dr. Satterthwaite,

We’re pleased to inform you that your manuscript has been judged scientifically suitable for publication and will be formally accepted for publication once it meets all outstanding technical requirements.

Kind regards,

Vitor Hugo Rodrigues Paiva, Ph.D.

Academic Editor

PLOS One
---

## [Editor Report · Acceptance letter]

PONE-D-25-51982R2

PLOS One

Dear Dr. Satterthwaite,

I'm pleased to inform you that your manuscript has been deemed suitable for publication in PLOS One. Congratulations! Your manuscript is now being handed over to our production team.

Kind regards,

on behalf of

Dr. Vitor Hugo Rodrigues Paiva

Academic Editor

PLOS One